# Corrections and Additions to Descriptions of Some Species of the Subgenus *Orthocladius* s. str. (Diptera, Chironomidae, Orthocladiinae)

**DOI:** 10.3390/insects13010051

**Published:** 2022-01-02

**Authors:** Bruno Rossaro, Laura Marziali, Giulia Magoga, Matteo Montagna, Angela Boggero

**Affiliations:** 1Dipartimento di Scienze Agrarie ed Ambientali, Università degli Studi di Milano, Via Celoria 2, 20133 Milano, Italy; giulia.magoga@unimi.it (G.M.); matteo.montagna@unimi.it (M.M.); 2CNR-IRSA Water Research Institute, National Research Council, Via del Mulino 19, 20861 Brugherio, Italy; laura.marziali@irsa.cnr.it; 3CNR-IRSA Water Research Institute, National Research Council, Corso Tonolli 50, 28922 Verbania Pallanza, Italy; angela.boggero@irsa.cnr.it

**Keywords:** chironomids, midges, taxonomy, morphometry, principal component analysis

## Abstract

**Simple Summary:**

The biodiversity study and conservation are primary objectives in preventing loss of species consequent to global climatic change. The present contribution adds information to a key taxon, a holometabolous insect belonging to Order Diptera, family Chironomidae, whose larvae are members of freshwater macroinvertebrate fauna and are considered good indicators of water quality.

**Abstract:**

The larvae of some species of the subgenus *Orthocladius* s. str. (Diptera, Chironomidae) are here described for the first time with corrections and additions to the descriptions of adult males and pupal exuviae. The identification of larvae is generally not possible without association with pupal exuviae and/or adult males, so the descriptions here are based only on reared material or on pupae with the associated larval exuviae. Usually, Chironomidae larvae can be separated on the basis of morphometric characters, the most discriminant ones are: (1) the ratio between the width of median tooth of mentum (Dm) and the width of the first lateral tooth (Dl) = mental ratio (DmDl), (2) the ratio between the length of the first antennal segment (A_1_) and the combined length of segments 2–5 (A_2–5_) = antennal ratio (AR). The shape of mandible, maxilla, and other body parts are almost identical in all the species considered in this study. The larva of *Orthocladius (Symposiocladius) lignicola* is very characteristic and can be separated by the shape of mentum and the larvae of all the known species of *Symposiocladius* are characterized by the presence of large Lauterborn organs on antennae and of tufts of setae on abdominal segments. The larvae of *Orthocladius (Orthocladius) oblidens* and *Orthocladius (Orthocladius) rhyacobius* can be distinguished from other species basing on their large Dm and to each other by AR. A principal component analysis was carried out using 5 characters: (1) Dm, (2) Dl, (3) length of A_1_, (4) width of A_1_ (A_1_W), (5) combined length of segments 2–5 (A_2–5_). The most discriminant characters were Dm and A_1_, confirming that DmDl and AR can be used to separate species at larval stage, but the large superposition of morphometric characters in different species confirms that association with pupal exuviae is in any case needed to identify larvae. In future perspective, the development of reference DNA barcodes from specimens identified by specialists is recommended since possibly the best tool for larvae identification, but association of barcodes with morphotypes is in any case fundamental.

## 1. Introduction

The genus *Orthocladius* Wulp, 1874 is one of the richest in species within the Chironomidae subfamily Orthocladiinae [1]. It is among the most difficult genera to morphologically identify, as larvae of many species are quite similar and almost impossible to be separated basing purely on morphological criteria. Pupal exuviae can be better separated, but some species pose problems also at the pupal stage. Adult males are well separated in many cases, notably relying on features of the hypopygium, yet some can be confounded also in the adult stage.

Within *Orthocladius*, taxonomic changes were repeatedly proposed for the generic/subgeneric division. For instance, the new genus *Symposiocladius* [2] was erected, but subsequently it was reduced to subgenus rank [3]. *Symposiocladius* was then revised with the description of new species [4], and more recently a new subgenus, *Mesorthocladius* [5], was erected.

At present, the genus *Orthocladius* includes 142 species [1], with the subgenus *Orthocladius* s. str. including at least 62 species. Most species are Palaearctic (104), many are Nearctic (54) with a few in the Afrotropical or Oriental regions [1]. Some species such as *Orthocladius (Orthocladius) decoratus, Orthocladius (Orthocladius) dentifer, Orthocladius (Orthocladius) oblidens, Orthocladius (Orthocladius) rubicundus, Orthocladius (Symposiocladius) holsatus* and *Orthocladius (Symposiocladius) lignicola* are Holarctic; other species are cited from both the Palaearctic and Nearctic regions, but they need to be checked for taxonomic identity. For example, *Orthocladius (Orthocladius) obumbratus*, originally described from the Nearctic and *Orthocladius (Orthocladius) excavatus*, originally described from the Palaearctic, were probably confused; in particular, the presence of *O. obumbratus* in the Palaearctic is questionable [6]. A list of species belonging to the subgenera *Mesorthocladius, Orthocladius, Pogonocladius* and *Symposiocladius* is given in Appendix A, with some nomenclature, taxonomic and distribution notes.

Until now, many contributions were published allowing the identification of some *Orthocladius* species at the different metamorphic stages (larvae, pupae, adults). In particular, the Nearctic species of the subgenus *Orthocladius* were revised including all stages for many species [7]; the West Palaearctic species were revised, including adult males and pupal exuviae [6]; pupal exuviae of many species of the West Palaearctic Region were described in detail and a key to species was given [8]. Moreover, adult males from England and Ireland were in depth investigated, figures and a key to species published [9].

In general, pupal exuviae can be considered the most useful stage for separating species, indeed some species have pupal exuviae as type material, such as *Orthocladius (Orthocladius) pedestris* and *Orthocladius (Orthocladius) rivinus* [10]. Most species can be identified also from adult males, although sometimes with more difficulties (see Keys). Larvae generally are not separable on the basis of morphological characters, but some morphometric details can be useful in some cases (see Results).

The aim of the present paper is to describe the larvae of some species of the subgenus *Orthocladius* s. str., for which larval, pupal exuviae and adult males are available from reared material. Corrections and additions to the already available descriptions of adult males and of pupal exuviae are also provided. Some species of the subgenera *Symposiocladius, Mesorthocladius* and *Pogonocladius* are also considered in the study because of their similarity with certain species of the subgenus *Orthocladius* s. str. In light of the last observation, the validity of the separation of some subgenera is discussed.

Keys to adult males and pupal exuviae are proposed, excluding the subgenera *Euorthocladius* and *Eudactylocladius*; characters useful to separate larvae are given, but dichotomous key to identify larvae cannot yet be proposed.

## 2. Materials and Methods

All the individuals sampled for this study were collected between 1974 and 2016. Larvae were collected with a Surber net, pupal exuviae and pupae with a Brundin net [11], adults with a hand net [9]. Larvae and mature pupae were transported alive to the laboratory using a portable refrigerator. Individuals were singularly reared to adults within Petri dishes or within glass tubes in a controlled-temperature chamber at a temperature ranging from 6 to 15 °C. Moreover, egg mass rearing was carried out within small tanks at the same temperature range mentioned before, in which dissolved oxygen saturation was guaranteed with aeration. A photoperiod of 14 h of light and 10 h of dark was maintained using a fluorescent lamp OSRAM LUMILUX COMBI-N/P, 18W.

Specimens from the collection of the Institut royal des Sciences naturelles de Belgique (IRSNB) and from the Swedish Museum of Natural History (NHM) were examined, including type material. These specimens, previously mounted on slides, dry pinned, or stored between two celluloid layers in isinglass, were mounted on permanent slides. Pinned specimens were prepared by boiling in KOH 10%, except the wings, transferred in acetic acid, butanol and in a phenol: xylene mixture 3:1, then mounted in balsam on a microscope slide. Specimens in isinglass were also gently boiled in KOH to dissolve gelatin, and thereafter treated as described above.

Sampled and reared individuals were fixed in 75% ethanol for preparation and mounted on microscopic slides according to Sæther [12] and Wirth & Marston [13], using balsam or Faure as mounting medium. In the case of successful rearing, adult, larval and pupal exuviae were mounted on the same slide. The adult abdomen (including genitalia) was mounted in a dorso-ventral position. To examine the virga at high magnification (1000–1250×), it was sometimes necessary to dissect the IX tergite and to mount it as a separate part.

Descriptions provided in this study are based only on reared adult males with associated pupal exuviae and larvae, when available. Body parts measurements were made at different magnifications (10–1000×) using a LEICA DM LS B2 optic microscope connected to a LEICA DFC320 camera and analysed with LEICA LAS software V4.8. Measurements were given in μm unless otherwise stated. Photos of characters of taxonomic interest were taken with the LEICA DFC320 camera.

Adopted terminology follows Soponis [7], Sæther [14] and Langton [15].

Morphometric data were analysed using R software version 4.1.2 [16]. Principal component analysis (PCA) was carried out using the R package vegan [17].

## 3. Results

### 3.1. Selected Species

A key for the identification of adult males and pupae of *Orthocladius* s. str. species and closest species is provided. The key does not include the subgenera *Euorthocladius* and *Eudactylocladius* since they can be easily separated from *Orthocladius* s. str. both as larvae, pupae and adults [8,9], whereas the species of the subgenera *Mesorthocladius, Pogonocladius* and *Symposiocladius* are included in the key, because their separation from *Orthocladius* s. str. species often is not straightforward. The list of the species included in the key is reported in Table 1, where asterisks (*) indicates that all the three life stages (male adults, pupal exuviae, larvae) belonging to the same specimen were examined, to guarantee membership to the same species.

### 3.2. Subgeneric Characters of Adult Male of *Orthocladius* s. str.

Body color. Head dark-yellow brown, antennal setae gold-dark brown, scapus black. Thorax: background thoracic color yellow-brown; mesonotal stripes, humeral region, metanotum and mesosternum darker than the background color. Abdomen, femur and tibiae dark, tarsomeres lighter, halteres from transparent to white (character not usable for mounted material, being observable only in pinned specimens). Body color is attenuated in specimens mounted in Canada balsam after 10% KOH clearing, but live or pinned specimens are generally not available for study, so only the color observable in slide-mounted specimens is considered and is of low utility.

Antenna. With 13 flagellomeres, plume well developed; 13° flagellomere not thickened to club, groove beginning at flagellomere 3–4, long sensilla chaetica present on flagellomeres 2–5 and 13; subapical seta absent at apex. Antennal ratio 1.1–2.6.

Head. Eyes bare, with a median projection. Temporals can be divided in 4–6 inner verticals, 4–6 outer verticals and 2–4 postorbitals. Palps with 5 segments (pm1–5), pm3 longer or subequal to pm4.

Thorax. Antepronotal lobes well developed, but narrowed medially, with a shallow median notch and a group of well developed lateral setae. Acrostichals begin near antepronotum. Dorsocentrals generally stout and long (ranging between 75–125 µm), in few cases (*O. rubicundus*) slender and shorter (less than 40 µm long), in one row, some setae not well lined up in some cases. Scutellars usually uniserial, not lined up or multiserial in *Orthocladius (Orthocladius) wetterensis*.

Wing. Wing membrane without setae, with fine punctation, microtrichia visible at 400×. Costa at most moderately extended beyond R_4+5_. R_2+3_ ending from 1/3 of distance from R_1_ to R_4+5_. R_4+5_ generally ending distal to end of M_3+4_, FCu little distal to RM or at the same level (VR 1–1.2). Cu1 slightly bent, squama with setae. R_1_ and R_4+5_ generally without setae. Anal lobe rounded to strongly produced.

Legs. Pseudospurs generally present on tarsomeres 1 and 2 of mid leg and tarsomere 1 of hind leg, sensilla chaetica sometime present on Ta_1_ of P_3_. Pulvilli absent, claw with 4 teeth apically.

Hypopygium. Anal point with lateral setae, usually triangular with pointed apex, with rounded apex only in subgenera *Euorthocladius* and *Mesorthocladius*. Virga present and well developed or very reduced to absent in subgenus *Symposiocladius* and in *Orthocladius (Orthocladius) rhyacobius*. Superior volsella hook-like, triangular or collar- like. Inferior volsella divided into a dorsal and a ventral lobe, dorsal lobe long and narrow, noselike, pinched, short and squared, short and rounded [7], differences have taxonomic value; IVr is here introduced as the ratio between the length of dorsal lobe and its width. Ventral lobe covered by dorsal lobe, well extended beyond dorsal lobe in some species. Gonostylus simple tapered to end or forming a right angle at outer margin, with a well developed megaseta, crista dorsalis present, but generally not developed, very developed only in a few species.

### 3.3. Subgeneric Characters of Pupal Exuvia of *Orthocladius* s. str.

Color in species mounted on slides variable from yellow-gold to brown-black. Pigmentation with different distribution on abdominal segments, posterior margin of abdominal segments much darker in some species. Frontal setae on small cephalic tubercles, frontal warts [14,15] present and more (*O. pedestris*) or less (*O. rubicundus*) developed. Thoracic horn usually with small spines, elongated (200–500 µm long), very long, slender and unarmed in *Orthocladius (Mesorthocladius) frigidus* and in *Orthocladius (Mesorthocladius) vaillanti* [18,19], enlarged in the middle in *O. wetterensis* and more or less rounded or pointed to apex in other species. The thoracic horn is sometimes bent at apex, but its shape is influenced by mounting position. Rows of hooks always present on posterior margin of tergite II (T_II_)_,_ width of hooks area variable in different species and of some taxonomic value: e.g., it is narrower in *O. oblidens* than in *O. rhyacobius*. Adhesion muscle-marks present on tergites: two oblique marks in the postero-median area, 2–5 less visible marks in the antero-lateral area on each side. Five dorsal setae (D_I–V_) are present on tergites, three ventral setae (S_I–III_) on sternites. Few setae on sternites may be branched as in *Orthocladius (Orthocladius) rivinus*. T_II-V_ with very small points forming an area (= point patch), which may be divided into an anterior, median, posterior and apical field, continuous (*O. rhyacobius*) or well separated from each other (*O. rubicundus*). T_VI_ with only an anterior, median and posterior field, T_VII–VIII_ without point fields. The fields of points are more or less developed and extended laterally, often very small points may be observed extending to the antero-lateral corner of abdominal tergites, beyond the adhesion marks. Point patches are present also on sternites (especially S_II–III_) with points often joined into groups of 2–3 (“Gruppen-shagreen”). The small points on the antero-lateral corner of tergites are variable in extension, similar points are present on sternites and in specimens mounted in dorso-ventral position dorsal and ventral points may be confounded. The number of antero-lateral adhesion marks and the position of seta D_I_ are variable and have no taxonomic value. The different size of anterior, median, posterior and apical field of points on tergites is emphasised as an easily observable character without substantial variation within species, and thus a good candidate to be used in taxonomic keys for separating species.

Usually, very small blotches are present at the base of apical points on tergites, but in one species (*O. pedestris*) there are large brown blotches extending laterally to the apical point area of tergites and postero-medially on sternites. Pedes spurii B are present but reduced on segment II in *O. frigidus* and absent in *Orthocladius (Symposiocladius) ruffoi*. Pedes spurii A are present on S_IV–VII_, reduced on S_VII_ in some species. Five, or more rarely four, lateral setae are present on abdominal segment VIII, with variation within species, very well developed in some species (e.g., *O. ruffoi*), sometime the anterior and/or the posterior one are bifid, with variation within species. Anal lobes with 3 anal macrosetae, usually strongly hooked at tip, but straight in *O. ruffoi,* straight or gently curved in *O. dentifer*. Tips of anal lobes with or without taeniate extensions of the cuticle. When present, they have different consistency and a taxonomic value, varying from very small, colourless light extensions (*O. excavatus*) to ‘broken’ setae (*O. rubicundus*), teeth [10], well sclerotized chitinous spurs (*O. rhyacobius, O. pedestris*) [6] or spiniferous processes (subgenus *Symposiocladius*) [4,20]. Additional small taeniate extensions, sometime appearing as a weak fringe of short setae, may be present (*O. excavatus*, *O. pedestris*). A well developed fringe of setae or hair-like teeth [10] is present only in *O. vaillanti* [6,19], a less developed one in *O. ruffoi* [6]. The ratio between the lengths of anal macrosetae and anal lobes varies around 1 and has taxonomic value.

### 3.4. Subgeneric Characters of Larva of *Orthocladius* s. str.

This description is based only on species reared to adults and included in the Keys to adult males and pupal exuviae. Medium sized larvae, up to 10 mm long.

Antenna with 5 segments, decreasing in size from 1 to 3, 4th segment equal or longer than 3rd, 5th segment small. Ring organ near the basis of 1st segment, blade shorter than flagellum, style long as 3rd segment. AR or the ratio between the length of the first segment and the other 2–5 has taxonomic value. Lauterborn organ developed only in larvae of the subgenus *Symposiocladius* (Figures 66–70 in [4]).

Labrum. S_I_ bifid, S_II–SIII_ simple. Labral lamellae absent. Pecten epipharyngis with 3 narrow spines. Premandible simple, without brush.

Mandible with a long apical tooth and 3 inner teeth increasing in size, the distal one (3rd) is the largest, a distal sclerotization of mola may be mistaken as a 4th inner tooth; seta subdentalis apically pointed, seta interna consisting of about 5–8 branches, some branch bifid, outer margin smooth, mola without spines.

Maxilla: chaetulae of palpiger triangular or leaflike, galea with simple lamellae anteriorly and several rows of pectinate lamellae dorsally, pecten galearis absent, seta maxillaris simple.

Mentum. With a single median tooth and 6 lateral teeth. Ventromental plates reduced. Setae submenti well developed, inserted just a little more distal to the end of ventromental plates, about 50–70 µm long. Median tooth (Dm) of variable extension, from 1 to 4 times larger than the first lateral tooth (Dl1); the ratio between the width of median tooth and the 1st lateral one (DmDl) has taxonomic values.

Abdominal segments without robust setae, a tuft of setae present only in larvae of Symposiocladius (Figure 5l in [20]. Procerci only a little longer than wide, with 5–7 apical setae 350–400 µm long

### 3.5. Description of Species

The species considered in the keys to adult males and pupal exuviae are in Table 1. In Appendix A there is a legend of additional information given (Appendix A). Hereafter are reported additional notes for some species. References to contributions to description of life stages present in previous publications and species distribution are given in Appendix A, with additional taxonomic notes. Morphometric measures of adult males of some species are summarized in Table 2. Morphometric measures of larvae are in Table 3. A detailed list of all larval measures is in Appendix A. Measures from larval exuviae reared to adults were preferred, when available (see * in Table 1), otherwise measures are from larvae only tentatively assigned to a species. Ranges of measures are given when available. When a value is given before range, it is the value considered more accurate, because it comes from the best mounted specimen, because some measures are affected by error bound to the different position of the mounted parts [21].

***Orthocladius (Mesorthocladius) frigidus*** (Zetterstedt, 1838)

The species was well described in all three stages [6,8,9,22].

***Orthocladius (Mesorthocladius) vaillanti*** Langton & Cranston, 1991

The species was well described in all three stages [6,8,10,18,19].

***Orthocladius*****(*Orthocladius*) *decoratus*** (Holmgren, 1869)

The species is described as adult male and pupal exuviae [6,8,23]. The larva is still undescribed, probably cannot be separated from the larva of *O. excavatus.* The measures here given are from larvae collected in Svalbard at 3 August 2003, gently furnished from Museo delle Scienze (TN, Italy).

AR = 1.45–2.11, DmDl = 1.27–2.06, A_1_ = 50 (43–62) long, 15.5 (13–17) wide, A_2–5_ = 30 (21–39), Dm = 19 (16–21), Dl = 11 (10–12).

***Orthocladius (Orthocladius) dentifer*** Brundin, 1947

Only the adult male [6,22] and pupal exuviae [6,8] are described, the larva is unknown.

***Orthocladius (Orthocladius) excavatus*** Brundin, 1947

The species is described in several papers [6,24,25]. The adult male of *O. excavatus* is separated from other species by the presence of a developed virga (Figure 1), the triangular superior volsella and the digitiform dorsal lobe of inferior volsella. The adult male may be confounded with *O. marchettii*, but this species has a larger dorsal lobe of inferior volsella.

The pupal exuviae are characterized by reduced taeniate extensions and extensive cover of points on abdominal tergites III to VI.

The larva has a low mentum ratio DmDl, and an AR between 1.8–1.9 (Figure 1i–l).

Description from reared larvae, 15 January 1981, Ticino River, Boffalora Ticino, Metropolitan City of Milan, Italy.

AR = 1.81–1.90, DmDl = 1.96–3.32, A_1_ = 52 (50–60) long, 17 (14–17) wide, A_2–5_ = 27 (27–33), Dm = 19 (19–21), Dl = 9.6 (9–12).

***Orthocladius (Orthocladius) glabripennis*** (Goetghebuer, 1921)

The species was described in [26] and redescribed in [27], the larva is still not described.

The measures are from a larva collected in Curone stream (LC, Italy) at 15 March 2017 reared to adult tentatively assigned to *O. glabripennis*. AR = 2.15, DmDl = 2.81, A_1_ = 63 long, 18 wide, A_2–5_ = 29, Dm = 30, Dl = 10.

***Orthocladius (Orthocladius) marchettii*** Rossaro & Prato, 1991

*O. marchettii* was described in [28] and redescribed in [6] as adult male and pupal exuviae.

The species can be separated from *O. excavatus* by the larger dorsal lobe of the inferior volsella (Figure 2c), and the more prominent superior volsella (Figure 2b).

The pupa is characterized by the absence of taeniate extensions (Figure 2h), but can be confounded with the pupa of *O. glabripennis*. Only the darker color of abdominal segments (Figure 2e) can separate these species.

The larva has a variable width of median tooth of mentum (Figure 2j).

Description from reared larva, 3 January 1990, Aterno River, L’Aquila, Italy.

AR = 1.95–2.03, DmDl = 2.05–2.56, A_1_ = 68.18 (60.37–68.18) long, 19.03 (15.31–19.03) wide, A_2–5_ = 34.02, mentum median tooth 28.63, Dl = 13.96 (13.79–14.13).

***Orthocladius (Orthocladius) majus*** Goetghebuer, 1942

The adult male was described in Langton & Pinder [9] briefly, and redescribed in [6] as adult and pupal exuviae. The pupal exuviae were also described in [8]. The larva is undescribed.

***Orthocladius (Orthocladius) nitidoscutellatus*** Lundström, 1915

The species is a senior synonym of *Orthocladius trigonolabis* Edwards, 1924 and was recently redescribed [23]. The larva is unknown.

***Orthocladius (Orthocladius) oblidens*** (Walker, 1856)

The adult male was described in [9,24,25], the pupal exuviae in [8,15]; the species was redescribed in [6] as adult male and pupal exuviae.

The adult male is separated by the large dorsal lobe of the inferior volsella (Figure 3d), the pupal exuviae are characterized by short anal macrosetae (Figure 3h). The larva by a large median mental tooth (Figure 3j).

Description from reared larva, 28 February 2004, Serio River, Ghisalba, Bergamo, Italy.

AR = 1.57 (1.57–2.54), DmDl = 4.63 (3.75–4.63), A_1_ = 54.63 (48–62) long, 23.42 (15–24) wide, A_2–5_ = 34.75 (21–35), Dm = 36.18 (32–37), Dl = 7.82 (7.64–8.00).

***Orthocladius*****(*Orthocladius*) *obumbratus*** (Johannsen, 1905)

This species was firstly described from Nearctic, its presence in the Palaearctic region was recently questioned [6]; the reason is that the lectotype of *O. obumbratus* differs from the specimens collected in other Nearctic stations and identified as *O. obumbratus* [7]. An accurate examination of dorsal and ventral lobe of inferior volsella from type material of *O. obumbratus* shows that the dorsal lobe is short and squared, near to the one of *O. oblidens*, with a length/width ratio (IVr) below 2 (Figure 4d). Other North American specimens, gently furnished from Dr. Caldwell and identified as *O. obumbratus,* have a digitiform inferior volsella (Figure 4h) with an IVr near to 3, a value observable in *O. excavatus*, which has always an IVr well above 2 (4 in the lectotype of *O. excavatus*). The examination of the lectotype and of the 2 paratypes of *O. obumbratus* should suggest that *O. obumbratus* is a junior synonym of *O. oblidens* and that the most of the material reported as *O. obumbratus* from Nearctic region may be a misidentification of *O. excavatus* [29]. An alternative hypothesis, here proposed, is that *O. obumbratus* could be a valid species somewhat intermediate between *O. oblidens* and *O. excavatus*, and the type material is somewhat atypical (!). If we accept this hypothesis the Palaearctic specimens identified as *O. obumbratus* can be considered as a misidentification of *O. excavatus*, while the specimens from Nearctic can still be assigned to *O. obumbratus*.

***Orthocladius (Orthocladius) pedestris*** Kieffer, 1909

The adult male of the species was described in [6,9], type material are pupal exuviae [10]. The adult is characterized by the hooked superior volsella (Figure 5b), and can be confounded with *O. decoratus*. The pupa is characterized by brown blotches in the intersegmental area of abdominal segments. The larva is very similar to those of *O. excavatus* and *O. marchettii* and is characterized by a low mental ratio DmDl and high AR (Figure 5i–l).

Description from reared larva, 23 April 2002, Taro River, Compiano, Parma, Italy.

AR = 2.23, DmDl = 2.19, A_1_ = 65.40 long, 16.94 wide, A_2–5_ = 29.37, Dm = 28.4, Dl = 12.98 (11.7–13).

***Orthocladius (Orthocladius) rhyacobius*** Kieffer, 1911

The species was described in [6,27] as adult male and its pupal exuviae in [6]. The adult is characterized by the absence or the strong reduction of virga (Figure 6). The lectotype was established based on a pupal exuvia [10]. *O. rhyacobius* and *O. excavatus* were considered junior synonyms of *O. obumbratus* in [10], but the pupal exuvia of *O. rhyacobius* has very strong taeniate extensions (Figure 6g), whereas *O. excavatus* has small taeniate extensions (Figure 1g). The associations of reared pupal exuviae with adult males support the evidence that *O. rhyacobius* and *O. excavatus* are different species, the former lacking virga (Figure 6b), the latter with a well developed virga (Figure 1b). The larva of *O. rhyacobius* is characterized by a large median mental tooth and a high AR (Figure 6i–l).

Description from reared larva, 18 December 1980, Po River, Caorso, Piacenza, Italy.

AR = 1.85–1.92, DmDl = 3.74–3.94, A_1_ = 54.42 (51.24–58.75) long, 16.84 (13.84–16.84) wide, A_2–5_ = 28.86 (25–32), Dm = 33.16 (26.32–34), Dl = 8.86 (7.96–9.53).

***Orthocladius (Orthocladius) rivinus*** Potthast, 1914

The lectotype was established on pupal exuviae [10]. The species was described in [6,9] as adult male. The adult is characterized by a triangular superior volsella (Figure 7c) and a squared dorsal lobe of inferior volsella (Figure 7d). Pupal exuviae are characterized by the splitting of some setae on sternites III (Figure 7g).

The larva has a median mental tooth and an AR with intermediate values within the subgenus so it is difficult to identify this species at the larval stage (Figure 7i–l).

Description from reared larva, 26 February 1990, Aterno River, L’Aquila, Italy.

AR = 1.75, DmDl = 2.64, A_1_ = 55.82 long, 15.98 wide, A_2–5_ = 31.91, Dm 25.07, Dl 9.49.

***Orthocladius (Orthocladius) rubicundus*** (Meigen, 1818)

The species was described as adult male [9,25] and as pupal exuviae [8,15] and redescribed in [6] from adult male and pupal exuviae.

The adult male (Figure 8) is characterized by the small and thin dorsocentral setae (see Section 3.6
*Key to adult males*).

The pupal exuviae are characterized by the presence of a postero-lateral patch of granules (chitinized rings) on apical bands of Tergites II–V extending ventrally, the patch of granules may be complete ventrally across the sternites or medially broken (Figure 8f,g).

The larva has a variable AR and a median mental tooth with intermediate values (Figure 8i–l).

Description from reared larva, 15 November 1978, Oglio River, Palazzolo, Brescia, Italy.

AR = 1.58–1.76, DmDl = 2.66–2.80, A_1_ = 45.39–52.13 long, 13.58–14.23 wide, A_2–5_ = 23.22–26.06, Dm = 23.56 (22.86–23.44), Dl = 8.39 (7.94–9.24).

***Orthocladius (Orthocladius) wetterensis*** Brundin, 1956

The adult male was described in [9,24], the pupal exuviae in [8] and redescribed in [6] from adult male and pupal exuviae. Inferior volsella is very characteristic (Figure 9d), scutellar setae are arranged in multiple rows (Figure 10d), this character separates *O. wetterensis* from other species. The pupal exuviae are characterized by an enlargement in the middle of the thoracic horn (Figure 9e). The larva is reported has having a high AR [30], but the AR is low in reared larvae, lower in comparison with other species, so this character cannot be used to identify the species. More useful is the width of median tooth of mentum (Figure 9j).

Description from reared larva, 21 January 2000, 9 December 2001, Elvo Stream, tributary of Cervo, Tributary of Sesia River, Biella, Italy.

AR = 1.34–1.40, DmDl = 3.16–3.24, A_1_ = 48.75 (46.18–51.55) long, 16.58 (16.47–16.75) wide, A_2–5_ = 34.93 (33.81–37.08), Dm = 31.71 (26.97–37.38), Dl = 9.77 (7.94–9.77).

***Orthocladius (Pogonocladius) consobrinus*** (Holmgren, 1869)

The adult male is described in [9,26], the pupal exuviae in [8]. Absence of sensilla chaetica in flagellomeres 2–5 cannot be used as a character separating the subgenus [3], because they are always observed in the specimens examined, including type material deposited at NHM. The larva is briefly described (Figure 9.53G in [22]).

***Orthocladius (Symposiocladius) holsatus*** Goetghebuer, 1937

The species was well described in all three stages (Figures 1, 2, 4 and 5 in [20]). Some morphometric measurements of adult males taken from samples belonging to Brundin’s collection (NHM) are: IV = 15–16 × 12–14, pm3 = 254–272, pm4 = 210–212, anal point = 33–52 × 9–17.

***Orthocladius (Symposiocladius) lignicola*** Kieffer, 1914

The adult male is well known for a long time (Figure 5 in [4]), some morphometric measurements taken from samples deposited in IRSNB are: IV = 36–37 × 23–25, pm3 = 81–116, pm4 = 70–77, anal point = 55 × 17. Descriptions of pupal exuviae are in [4,8]. The larva was also described [2,4]; some pictures of larva are here added (Figure 11e–h).

***Orthocladius (Symposiocladius) ruffoi*** Rossaro & Prato, 1991

The pupal exuviae was described a long time ago sub *Rheorthocladius* sp. A [31], while the adult male was only recently described [6,9,27], (Figure 10e–h). Morphometric measures taken from samples deposited in IRSNB are: IV = 25–50 × 17–22, pm3 = 139–151, pm4 = 116–140, anal point = 38–43 × 13–12. Description of the type is in [28]. The larva is still undescribed.

### 3.6. Key to Adult Males of *Orthocladius* s. str.

1Anal lobe of wing strongly produced (Figure 9.50C in [3]), fore tarsus with a beard of long setae, AR > 2, inferior volsella finger like (Figure 22 in [6], Figure 170A in [9], Figure 37 in [24])
*consobrinus*

- with other combination of characters22Anal point very broad with rounded apex3
- anal point pointed at apex43virga present even if reduced, inferior volsella slender, crista dorsalis well developed (Figure 8, Figure 174C in [9], Figure 39 in [24])
*frigidus*

- virga absent, inferior volsella wider, crista dorsalis evident but not large (Figure 34 in [6,19])
*vaillanti*
4virga absent or very reduced, SV collarlike (Figure 6b,c and Figure 10f,g)5
- virga present (Figure 2b, Figure 3b, Figure 4b, Figure 5b and Figure 10a), not reduced, SV variable85IV digitiform, slender, IVr 2–3, AR 1.6–1.8, virga absent or very reduced (Figure 6d, Figure 23 in [6], Figure 41 in [24,27])
*rhyacobius*

- IV squared (Figure 10h), IVr < 1–2663rd/4th palpomere 1.10–1.16, AR 1.43–1.69, IVr low (Figures 1 and 2 in [20], Figure 172C in [9])
*holsatus*

- 3rd/4th palpomere, AR and IVr higher (Figure 10h)773rd/4th palpomere 1.20, AR 2.9 (Figure 30 in [6], Figure 171D in [9,28]) 
*ruffoi*

- 3rd/4th palpomere 1.50, AR 1.73 (Figures 1–4 in [2,4], Figure 43 in [7], Figure 172A in [9], Figure 71 in [25])
*lignicola*
8dorsocentrals reduced (Figure 10c), 20–40 µm long (Figure 10c), superior volsella triangular (Figure 8, Figure 28 in [6], Figure 171B in [9], Figure 70 in [25])
*rubicundus*

- dorsocentrals strong > 100 µm long, superior volsella collarlike, triangular or hooked99gonostylus triangular, with a very developed projection (Figure 32 in [6], Figure 49 in [7], Figure 72a in [25], Figure 51 in [23])
*nitidoscutellatus*

- gonostylus without a developed external spur, at most with rounded or toothed crista dorsalis1010gonostylus with a rounded very developed crista dorsalis (Figure 3 in [6], Figure 42 in [24], Figure 172D in [9])
*dentifer*

- gonostylus with a tooth shaped crista dorsalis, not appearing as an external spur (Figure 2d and Figure 5d)1111scutellars multirowed (Figure 10d, Figure 37 in [6], Figure 172B in [9], Figure 69 in [25,28])
*wetterensis*

- scutellars unirowed1212superior volsella collarlike (Figure 10 in [6])13
- superior volsella triangular or hooked1413gonostylus with a well developed angle on the dorsal side (Figure 10 in [6], Figure 173B in [9,25,26])
*glabripennis*

- gonostylus slender at apex without an evident crista dorsalis (Figure 12 in [6], Figure 171C in [9])
*maius*
14superior volsella hooked (Figure 5b)15
- superior volsella triangular (Figure 1c)1615IVr about 1.5 (Figure 1 in [6,23], Figure 68 in [25])
*decoratus*

- IVr > 3 (Figure 5c, Figure 20 in [6], Figure 173A in [9])
*pedestris*
16dorsal lobe of inferior volsella squared, IVr < 2.5 (Figure 3d and Figure 7d)17
- dorsal lobe dorsal lobe of inferior volsella elongated, IVr > 2.5 (Figure 1d and Figure 2c)1817IVr about 2 (Figure 3d and Figure 4d, Figure 17 in [6], Figure 173C,D in [9], Figure 40 in [24], Figure 66 in [25])
*oblidens*

- IVr about 1 (Figure 7d, Figure 26 in [6], Figure 174A in [9,28])
*rivinus*
18AR < 2, superior volsella more rounded, IVr about 3 (Figure 1c,d, Figures 5 and 6 in [6], Figure 38 in [24], Figure 67 in [25])
*excavatus*

- AR > 2, superior volsella more prominent, IVr about 2.5 (Figure 2b,c, Figure 14 in [6,28])
*marchettii*


### 3.7. Key to Pupal Exuviae of *Orthocladius* s. str.

1tergite II without posterior hook row, tergites II–VII with a pair of circular point patches, taeniate extensions absent (Figure 9.41G in [8,18])
*consobrinus*

- hook row present on tergite II, median point patches continuous on an extended area, taeniate extensions present or absent22thoracic horn thin walled, fragile, sinuous, long and narrow (Figures 6 and 7 in [19])3
- thoracic horn shorter, straight, less than 300 µm long and more than 20 µm wide43anal lobes without lateral setae, thoracic horn long, smooth and narrow, brown at base, colourless thereafter (Figure 9 in [6,8], Figure 7 in [19])
*frigidus*

- anal lobes with a fringe of setae (Figures 35 and 36 in [6], Figure 6 in [19])
*vaillanti*
4anal lobes with a spur, pedes spurii B developed on tergite II, absent on T_III_, anal lobes without fringe; anal macrosetae long, flexible and tapering to their tips (Figures 10–12 in [2], Figures 36, 42, 48 and 49 in [4])
*lignicola*

- anal lobes without a spur, taeniate extensions present or absent55median and posterior band of tergites II–VI transverse, that is well separated by an area free of points6
- median point patches continuous with posterior ones, even if circular area devoid of points may be present96granulations (rows of chitin rings) are present on the tergites laterally on the apical band (Figure 8f), a posterior band of granulations (still more conspicuous) is present on sternites too, often in a dark field (Figure 8g, Figure 29 in [6,8])
*rubicundus*

- chitin rings absent on posterior margin of abdominal segments77pedes spurii B present on T_II_ and T_III_. tergites II–V with median point patch transverse, separate from the posterior band of narrow, sharp points, differing markedly from the short, sturdy points of the posterior band, anal macrosetae straight or somewhat curved (Figure 56 in [4], Figure 4 in [20])
*holsatus*

- pedes spurii B absent or present only on segment II88pedes spurii B absent, anal macrosetae longer, 300–350 µm long, TVII with 5 robust lateral setae (Figure 11a–d), Figure 31 in [6])
*ruffoi*

- pedes spurii B present on segment II (Figure 4 in [6,8])
*dentifer*
9thoracic horn with a characteristic bulge in its middle (Figure 9e), points of posterior transverse band smaller than points of anterior and apical band (Figure 9f), anal lobes without taeniate extensions (Figure 38 in [6,8])
*wetterensis*

- thoracic horn without median bulge, points of posterior transverse band similar or larger than points of anterior bands, taeniate extensions often present1010taeniate extensions absent (Figure 2)11
- taeniate extensions present (Figure 1)1311abdomen segments dark brown (Figure 2, Figures 15 and 16 in [6,28])
*marchettii*

- abdomen segments colourless1212thoracic horn about 400 µm long, (Figure 11 in [6,8])
*glabripennis*

- thoracic horn smaller, at most 200 µm long, genital sacs very long (Figure 33 in [6,8,23])
*nitidoscutellatus*
13anal macrosetae short about 100–150 µm, at most curved, but not hooked at tips (Figure 3h); posterior band of T_III_ extending lateral to apical band (Figure 3f), anal lobes with strong taeniate extensions (Figure 3g, Figure 18 in [6,8])
*oblidens*

- anal macrosetae hooked at tips1414sternites II with branched setae (Figure 7g), posterior band of T_III_ extending lateral to apical band (Figure 3f and Figure 7f), Figure 27 in [6,8])
*rivinus*

- setae of sternite II simple, posterior band of tergite II not extended lateral to apical band1515large brown blotches extending laterally to the apical point area of tergites and postero- medially on sternites (Figure 5g, Figure 21 in [6,8])
*pedestris*

- brown blotches absent1616taeniate extensions well developed, with 3–4 spines 30–45 µm long, occupying an area of about 400 µm^2^ (Figure 6g, Figures 24 and 25 in [6])17
- taeniate extensions present but reduced, at most 30 µm long. Anal lobes may be weakly fringed with hair-like teeth in addition to the apical teeth (Figure 1g, Figure 7 in [6,8])
*excavatus*
17points on tergites and spines on thoracic horn strong (Figure 6e,f, Figures 24 and 25 in [6,8])
*rhyacobius*

- points on tergites and spines on thoracic horn less strong (Figure 3 in [6,23])
*decoratus*


### 3.8. Morphometric Measures of Larvae of *Orthocladius* s. str.

The characters used to identify the larvae of *Orthocladius* are generally able to separate subgenera [21,30,31,32]. Attempts to separate species within the subgenera are scanty [29,31]. The reason is that most species have very similar larvae. Here an attempt is made to separate species according to 7 morphometric characters (Table 3): only values measured on reared larvae are given.

Other characters as the length of each of the antennal segments 2–5, mandible teeth length, distance between setae submenti, size of head capsule [30] were no more considered because their measures are subjected to large errors.

A principal component analysis (PCA) was carried out using 5 selected characters to find the ones responsible of the largest variance (Table 4 and Table 5, Figure 12). The width of the median tooth of mentum (Dm) and the length of the first antennal segment (A_1_) were the most discriminating characters, while the width of the first antennal segment (A_1_w) was a little more discriminating than the length of antennal segments 2–5 combined (A_2–5_). In PCA, the width of the first antennal segment (A_1_W) was a little more discriminating than the length of A_2–5_ combined in the first axis, but the reverse was true in the second axis (Table 4). As a practical measure to separate species, the ratio DmDl (ratio between the width of the median and first lateral mental tooth) and the antennal ratio AR (ratio between the length of the first and the 2–5 length of antennal segments) were selected (Table 3, Figure 13). In the impossibility to give a dichotomous key rough outlines to separate species are given (Table 6).

## 4. Discussion

Species identification within the subgenus *Orthocladius* takes advantage from characters expressed in pupal exuviae. Most species can be distinguished also on the basis of differences in adult male genitalia or some other character as dorsocentral and scutellar setae, while others require the association of pupal exuviae with adult males for achieving identification (see Key). Absence of virga, collar-like superior volsella, ventral part of inferior volsella not extending below dorsal part were considered as characters supporting the separation of subgenus *Symposiocladius* from *Orthocladius* s. str. [4], but *O. rhyacobius* (Figure 6b) and O. *dentifer* (Figure 3 in [6])*,* not included in *Symposiocladius,* have characters compatible with this definition of the subgenus. The subgenus *Symposiocladius* include pupal exuviae with quite different morphological characters, i.e., both species with a large spur on anal lobes, such as *O. (S). lignicola* (Figures 10 and 12 in [2])*,* and species without spur, such as *O. (S). holsatus* (Figure 4 in [20]). The larva of *O. (S). lignicola* is quite different from the others of this subgenus, because of the peculiar mentum (Figure 19 in [2], Figure 76 in [4]), but the presence of large Lauterbon’s organs on antennae and of setal tufts on the abdominal segments of larvae of all the known species of *Symposiocladius* (Figures 13–19 in [2], Figures 60–70 in [4], Figure 5 in [20]) supports the separation of the subgenus. Contrasting evidence about the subgeneric limits of *Orthocladius* s. str. and *Symposiocladius* should be clarified.

After an accurate morphometric analysis of reared material, it is possible to conclude that in principle larvae cannot be identified without association with pupal exuviae. In some situation the larvae can be also confounded with species belonging to other related genera, as is the case of *Symposiocladius* larvae, which can be assigned to some species of *Cricotopus* (see point 104/103 at page 197 in [22]). Similarly, the subgenus *Cricotopus (Paratrichocladius)* has larvae which cannot be easily separated from species of *Orthocladius* s. str. (see point 112/113 at page 198 [22]). The morphometric characters suggested to separate larvae (AR and DmDl) are only indicative. Antennal ratio (AR) is useful to separate species, but the ratio between the width of median tooth and the 1st lateral tooth of mentum (DmDl) is probably the most useful character for separating larvae. In particular, *O. oblidens* and *O. rhyacobius* are separated from other species thanks to wider Dm (Figure 3j, Figure 6j, Figure 12 and Figure 13). Unfortunately, morphometric measures of larvae have large superposition in different species, e.g., in *O. rubicundus* and *O. excavatus*, and for some species (*O. marchettii, O. pedestris, O. glabripennis*) the data available are very limited. The criteria given in previous papers [30,32] are not supported by the present evidence.

DNA-based taxonomy could be very useful to identify species at each life stage, overcoming difficulties related to morphological identification. However, the development of reference DNA sequences allowing the identification of the species of *Orthocladius* is still in an embryonic phase. In fact, DNA barcodes are available for a few species (see Appendix A, [23,33]). Future needs include collaboration between morphological and molecular taxonomists for the development of barcode libraries for the molecular identification including species not barcoded yet [34].

*Orthocladius* taxonomy would benefit from bringing to light the specialized literature about the species described in local or less known journals. Very similar species, and possibly some of them conspecific, were described from Nearctic [7]. In the present paper one of these cases is discussed, i.e., the problem of the relation between *O. obumbratus* with *O. excavatus*, but further cases of similarity between Nearctic and Palaearctic species are known. Just to give another example, the similarity of *O. carlatus* (Roback, 1957) and *O. curtiseta* Sæther, 1969 with *O. rubicundus* would deserve further study.

Some *Orthocladius* s. str. species are known from the East Palaearctic [35,36,37], many of them described in all three stages (larva, pupa, adult male) with possible affinities with *O. excavatus* and *O. rubicundus.* Other *Orthocladius* species for which all the life stages have been described are known for Japan [38,39,40]; some are very similar to West Palaearctic ones. Here again, the problem of possible synonymy could be solved with the support of molecular taxonomy, even if finding specimens could be not easy due to very large number of species described and the distances among collection localities.

## Figures and Tables

**Figure 1 insects-13-00051-f001:**
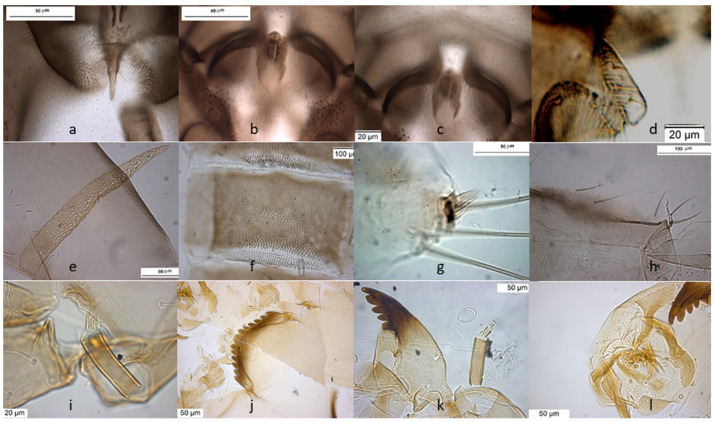
Morphological characters of *O.* (*O.*) *excavatus* adult male: (**a**) anal point; (**b**) virga; (**c**) superior volsella; (**d**) inferior volsella; pupal exuviae: (**e**) thoracic horn, (**f**) tergite III; (**g**) taeniate extensions; (**h**) tergite VIII; larva: (**i**) antenna; (**j**) mentum; (**k**) mandible; (**l**) labrum.

**Figure 2 insects-13-00051-f002:**
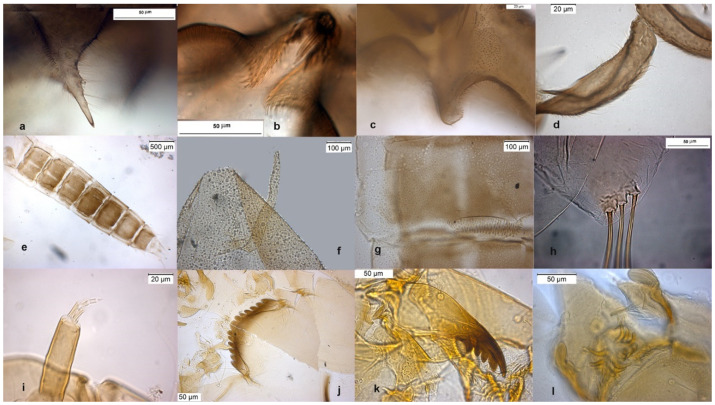
Morphological characters of *O.* (*O.*) *marchettii* adult male: (**a**) anal point; (**b**) virga and superior volsella; (**c**) inferior volsella; (**d**) gonostylus; pupal exuvia: (**e**) abdomen; (**f**) thoracic horn, (**g**) tergite II; (**h**) taeniate extensions; larva: (**i**) antenna; (**j**) mentum; (**k**) mandible; (**l**) labrum.

**Figure 3 insects-13-00051-f003:**
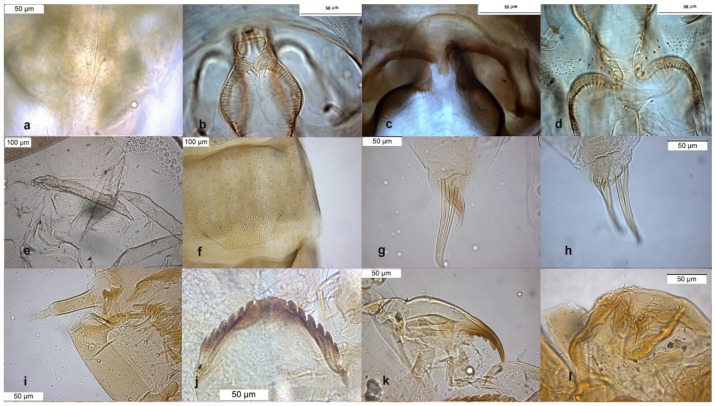
Morphological characters of *O. oblidens* adult male: (**a**) anal point; (**b**) virga; (**c**) superior volsella; (**d**) inferior volsella; pupal exuvia: (**e**) thoracic horn, (**f**) tergite III; (**g**) taeniate extensions; (**h**) anal macrosetae; larva: (**i**) antenna; (**j**) mentum; (**k**) mandible; (**l**) labrum.

**Figure 4 insects-13-00051-f004:**
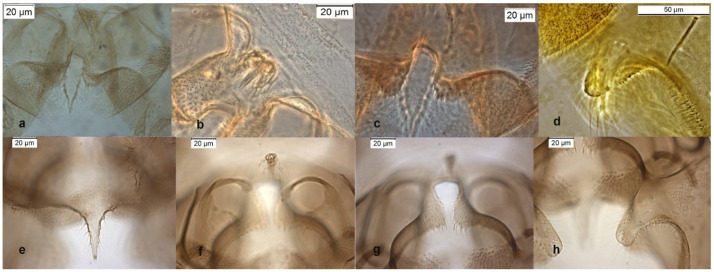
Morphological characters of *O. obumbratus* adult male lectotype: (**a**) anal point; (**b**) virga; (**c**) superior volsella; (**d**) inferior volsella; adult male Holly Ck USA-GA: (**e**) anal point; (**f**) virga; (**g**) superior volsella; (**h**) inferior volsella.

**Figure 5 insects-13-00051-f005:**
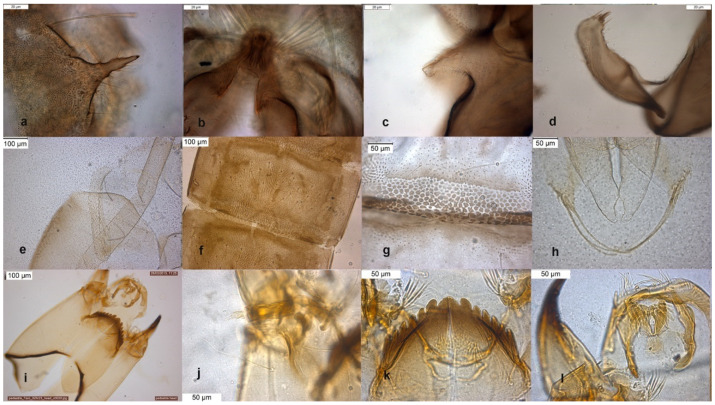
Morphological characters of *O. pedestris* adult male: (**a**) anal point; (**b**) virga and superior volsella; (**c**) inferior volsella; (**d**) gonostylus; pupal exuvia: (**e**) thoracic horn, (**f**) tergite IV; (**g**) sternite IV; (**h**) anal macrosetae; larva: (**i**) head; (**j**) antenna; (**k**) mentum; (**l**) labrum.

**Figure 6 insects-13-00051-f006:**
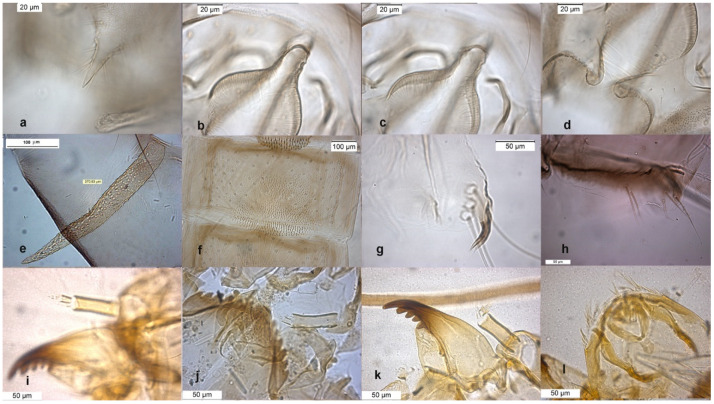
Morphological characters of *O. rhyacobius* adult male: (**a**) anal point; (**b**) virga; (**c**) superior volsella; (**d**) inferior volsella; pupal exuvia: (**e**) thoracic horn, (**f**) tergite III; (**g**) taeniate extensions; (**h**) tergite VIII; larva: (**i**) antenna; (**j**) mentum; (**k**) mandible; (**l**) labrum.

**Figure 7 insects-13-00051-f007:**
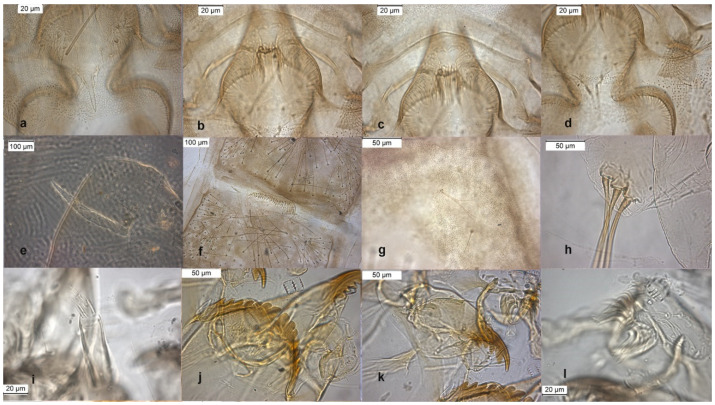
Morphological characters of *O. rivinus* adult male: (**a**) anal point; (**b**) virga; (**c**) superior volsella; (**d**) inferior volsella; pupal exuvia: (**e**) thoracic horn, (**f**) tergite II; (**g**) sternite III; (**h**) taeniate extensions; larva: (**i**) antenna; (**j**) mentum; (**k**) mandible; (**l**) labrum.

**Figure 8 insects-13-00051-f008:**
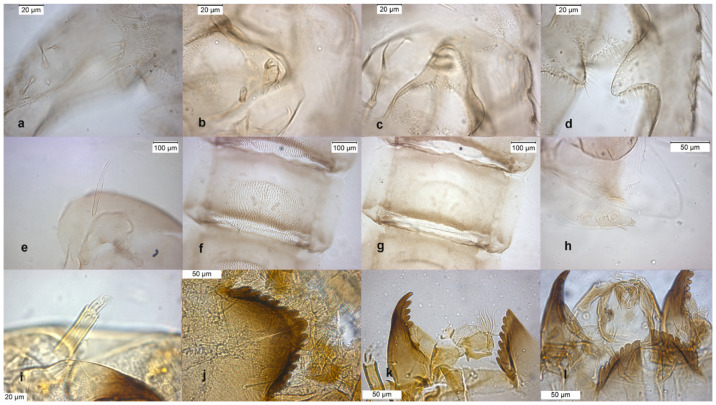
Morphological characters of *O. rubicundus* adult male: (**a**) anal point; (**b**) virga; (**c**) superior volsella; (**d**) inferior volsella; pupal exuvia: (**e**) thoracic horn, (**f**) tergite IV; (**g**) sternite-IV; (**h**) taeniate extensions; larva: (**i**) antenna; (**j**) mentum; (**k**) mandible; (**l**) labrum.

**Figure 9 insects-13-00051-f009:**
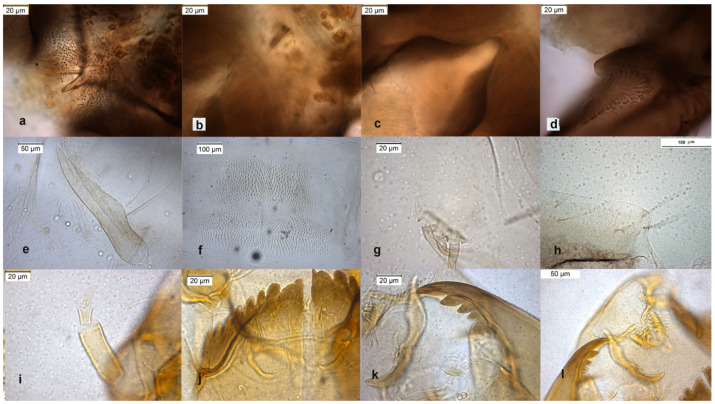
Morphological characters of *O. wetterensis* adult male: (**a**) anal point; (**b**) virga; (**c**) superior volsella; (**d**) inferior volsella; pupal exuviae: (**e**) thoracic horn, (**f**) tergite IV; (**g**) taeniate extensions; (**h**) TVIII; larva: (**i**) antenna; (**j**) mentum; (**k**) mandible; (**l**) labrum.

**Figure 10 insects-13-00051-f010:**
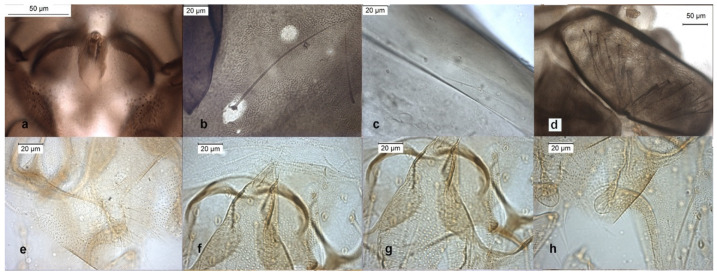
Morphological characters of adult males: (**a**) *O. excavatus* virga, (**b**) *O. excavatus* dorsocentral setae, (**c**) *O. rubicundus* dorsocentral setae, (**d**) *O.O. wetterensis* scutellum, (**e**) *O.S. ruffoi* anal point, (**f**) *O.S. ruffoi* virga, (**g**) *O.S. ruffoi* superior volsella, (**h**) *O.S. ruffoi* inferior volsella.

**Figure 11 insects-13-00051-f011:**
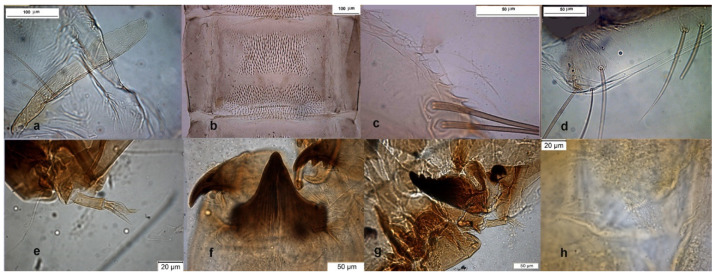
Morphological characters of *O.S. ruffoi* pupal exuviae: (**a**) thoracic horn, (**b**) tergite IV; (**c**) taeniate extensions; (**d**) tergite VIII. Morphological characters of *O.S. lignicola* larva: (**e**) antenna; (**f**) mentum; (**g**) mandible; (**h**) setal tufts.

**Figure 12 insects-13-00051-f012:**
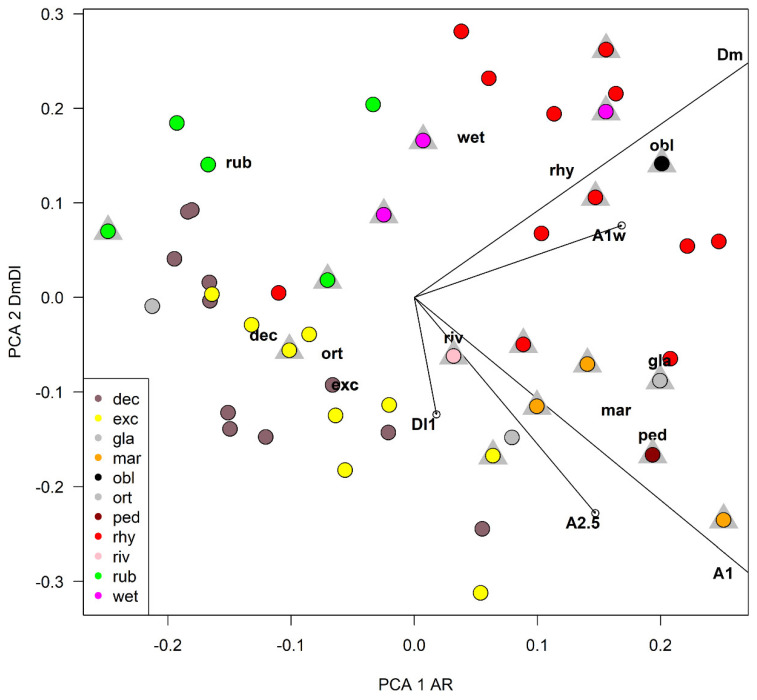
Plot of species scores and character loadings in principal component analysis in the first two axes, grey triangles refer to samples of larvae belonging to reared specimens, see Table 3 for abbreviations.

**Figure 13 insects-13-00051-f013:**
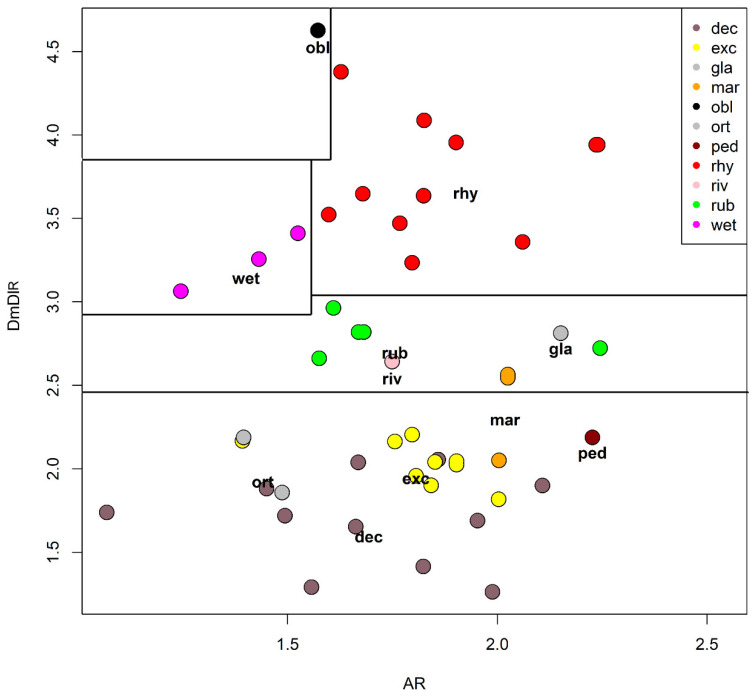
Plot of the ratio DmDl against the antennal ratio (AR). Dashed lines separates species with different morphometry, see Table 3 for abbreviations.

**Table 1 insects-13-00051-t001:** List of the species considered in the key.

(*) *Orthocladius* (*Mesorthocladius*) *frigidus* (Zetterstedt, 1838)
*Orthocladius* (*Mesorthocladius*) *vaillanti* Langton & Cranston, 1991
*Orthocladius* (*Orthocladius*) *decoratus* (Holmgren, 1869)
*Orthocladius* (*Orthocladius*) *dentifer* Brundin, 1947
(*) *Orthocladius* (*Orthocladius*) *excavatus* Brundin, 1947
*Orthocladius* (*Orthocladius*) *glabripennis* (Goetghebuer, 1921)
*Orthocladius* (*Orthocladius*) *maius* Goetghebuer, 1942
(*) *Orthocladius* (*Orthocladius*) *marchettii* Rossaro & Prato, 1991
(*) *Orthocladius* (*Orthocladius*) *oblidens* (Walker, 1856)
(*) *Orthocladius* (*Orthocladius*) *pedestris* Kieffer, 1909
(*) *Orthocladius* (*Orthocladius*) *rhyacobius* Kieffer, 1911
(*) *Orthocladius* (*Orthocladius*) *rivinus* Potthast, 1914
(*) *Orthocladius* (*Orthocladius*) *rubicundus* (Meigen, 1818)
*Orthocladius* (*Orthocladius*) *nitidoscutellatus* Lundström, 1915
(*) *Orthocladius* (*Orthocladius*) *wetterensis* Brundin, 1956
*Orthocladius* (*Pogonocladius*) *consobrinus* (Holmgren, 1869)
*Orthocladius* (*Symposiocladius*) *holsatus* Goetghebuer, 1937
*Orthocladius* (*Symposiocladius*) *lignicola* Kieffer, 1914
*Orthocladius* (*Symposiocladius*) *ruffoi* Rossaro & Prato, 1991

Asterisks (*) indicates that all the three life stages (male adults, pupal exuviae, larvae) belonging to the same specimen were examined, to guarantee membership to the same species.

**Table 2 insects-13-00051-t002:** Morphometric measures of adult males belonging to different species: wingL: wing length, AR: antennal ratio of adult male, IVr: ratio between length and width of dorsal lobe of inferior volsella.

Species	Abbreviation	wingL	AR	IVr
*O. decoratus*	dec	2.9	1.90	1.66
*O. dentifer*	den	3.2–3.5	1.85	1.46
*O. excavatus*	exc	2.7–3.0	1.93	3.01
*O. glabripennis*	gla	3.5–3.6	2.30	1.64
*O. marchettii*	mar	3.50	2.23	2.66
*O. oblidens*	obl	2.5–3.3	2.02	1.73
*O. obumbratus*	obu	2.70	1.77	2.19
*O. pedestris*	ped	2.23	1.60	3.55
*O. rhyacobius*	rhy	2.7	1.75	2.94
*O. rivinus*	riv	2.8	2.16	1.10
*O. rubicundus*	rub	2.1–2.3	1.45	2.00
*O. wetterensis*	wet	2.5	1.09	1.92

**Table 3 insects-13-00051-t003:** Morphometric measures of larval characters: A_1_ = length of first antennal flagellomere, A_2–5_ = combined length of flagellomeres 2–5, A_1_W = width of the first antennal segment, Dm = width of median tooth of mentum, Dl = wide of first lateral tooth, AR = antennal ratio A_1_/A_2–5_, DmDl = ratio of Dm to Dl.

	A_1_	A_2–5_	A_1_W	Dm	Dl	AR	DmDl
dec	50.27	30.28	15.50	19.24	11.47	1.69	1.70
exc	54.62	30.33	14.83	20.31	10.03	1.81	2.04
gla	63.27	29.41	18.72	30.39	10.81	2.15	2.81
mar	62.97	31.21	18.82	27.64	11.74	2.02	2.39
obl	54.63	34.75	23.42	36.18	7.82	1.57	4.63
ort	51.55	35.94	14.45	21.27	10.43	1.44	2.02
ped	65.40	29.37	16.94	28.40	12.98	2.23	2.19
rhy	54.42	28.86	16.84	33.16	8.86	1.92	3.74
riv	55.82	31.91	15.98	25.07	9.49	1.75	2.64
rub	45.39	26.06	13.58	23.56	8.39	1.76	2.80
wet	48.75	34.93	16.58	31.71	9.77	1.40	3.24

**Table 4 insects-13-00051-t004:** PCA analysis results, eigenvalues of the five axes and proportion of variance explained.

Eigenvalues		PC1	PC2	PC3	PC4	PC5
Eigenvalue		57.7669	34.8197	16.4581	3.4536	1.86469
Proportion	Explained	0.5051	0.3045	0.1439	0.0302	0.01631
Cumulative	Proportion	0.5051	0.8096	0.9535	0.9837	1

**Table 5 insects-13-00051-t005:** Factor loadings of the variables included in PCA.

Character	PC1	PC2	PC3	PC4	PC5
A_1_	3.9935	−3.3193	−0.8665	−0.17252	0.04569
A_2–5_	0.8999	−1.0826	3.1198	0.07901	0.11392
A1w	1.0303	0.3614	−0.3345	1.31736	0.46594
Dm	4.4264	3.1453	0.2221	−0.18348	−0.14844
Dl1	0.1106	−0.5872	0.13	0.65751	−0.97665

**Table 6 insects-13-00051-t006:** DmDl and AR values useful to separate larvae.

DmDl and AR Values	Species Epithet
DmDl > 3.5 and AR < 1.6 (Figure 3)	*oblidens*
DmDl > 3 and AR > 1.6 (Figure 6)	*rhyacobius*
DmDl 3–3.5 and AR < 1.6 (Figure 9)	*wetterensis*
DmDl 2.5–3 (Figure 8)	*rubicundus*
DmDl 2.5–3 and AR 1.7–1.9 (Figure 7)	*rivinus*
DmDl 2.5–3 and AR > 2.1	*glabripennis*
DmDl 2–3 and AR 1.9–2.1 (Figure 2)	*marchettii*
DmDl < 2.5 and AR < 2.1 (Figure 1)	*excavatus*
DmDl < 2.5 and AR > 2.1 (Figure 5)	*pedestris*
DmDl < 2.5	*decoratus*

## Data Availability

All specimens analyzed and the related data are deposited at University of Milan, if not specified otherwise, and can be requested to the corresponding author.

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
