# Peer review of "Corrections and Additions to Descriptions of Some Species of the Subgenus Orthocladius s. str. (Diptera, Chironomidae, Orthocladiinae)"

_insects, 2022, doi:10.3390/insects13010051_

Round 1

Reviewer 1 Report

The genus Orthocladius is one of the most difficult to identify in the subfamily. In most cases, it is possible to determine the species only in the presence of an imago, pupa and larva. But there are very few such works devoted to the study of all stages of the development of Orthocladius. Therefore, this study is in demand both by taxonomists and hydrobiologists, who have to determine only larvae and pupae in zoobenthos samples. Also, I believe this work is the basis for further study of the genus with using molecular genetic methods. I believe that the article should be published as soon as possible.

Author Response

Many thanks for having supported the manuscript with your authoritative opinion

Reviewer 2 Report

This an interesting manuscript dealing with the a complicated group, the Orthocladius. Although, the manuscript has an unusual format, mostly reporting to others previous studies, in my opinion, it can be published as it is.   

Author Response

I agree with the reviewer about the large use of cited literature, indeed the genus was very intensively studied in the past, but I thought that it was useful to update the large information available, fixing some still opened question   

Reviewer 3 Report

The manuscript is written consistently, providing an update on some subgenera of the Orthocladius genus. The morphometric analysis is original and well founded, giving interesting results. There is only one observation I wish to make, and it is the following. It is well known among experts in the family Chironomidae that the larvae of Orthocladius, and particularly the subgenus Orthocladius, are very difficult to determine in certain cases with those of the genus Cricotopus. I suggest making a comment about it, mentioning those situations where this problem can be found (e.g. Cricotopus and O. annectens and O. lignicola), both sharing setal tufts).

Author Response

I have included a short sentence in the Discussion (in yellow) were I add a comment about the relation of Orthocladius s. str. with related genera/subgenera Cricotopus and Paratrichocladius, in this sentence I cite the Cranston's at al. key, where the authors include the characters that can be used to separate species of different genera; I think that it was not the case to reproduce point to point the key. In any case the reviewer's  observation is very opportune.